# Functions of TAM Receptors and Ligands Protein S and Gas6 in Atherosclerosis and Cardiovascular Disease

**DOI:** 10.3390/ijms252312736

**Published:** 2024-11-27

**Authors:** Teagan Prouse, Samarpan Majumder, Rinku Majumder

**Affiliations:** 1Department of Interdisciplinary Oncology, Louisiana State University Health Sciences Center, New Orleans, LA 70112, USA; tprous@lsuhsc.edu; 2Department of Genetics, Louisiana State University Health Sciences Center, New Orleans, LA 70112, USA; smaju1@lsuhsc.edu

**Keywords:** Protein S, Gas6, atherosclerosis, TAM receptors, cardiovascular disease, myocardial infarction, apoptosis

## Abstract

Atherosclerosis and cardiovascular disease are associated with high morbidity and mortality in industrialized nations. The Tyro3, Axl, and Mer (TAM) family of receptor tyrosine kinases is involved in the amplification or resolution of atherosclerosis pathology and other cardiovascular pathology. The ligands of these receptors, Protein S (PS) and growth arrest specific protein 6 (Gas6), are essential for TAM receptor functions in the amplification and resolution of atherosclerosis. The Axl-Gas6 interaction has various effects on cardiovascular disease. Mer and PS dampen inflammation, thereby protecting against atherosclerosis progression. Tyro3, the least studied TAM receptor in cardiovascular disease, appears to protect against fibrosis in post-myocardial infarction injury. Ultimately, PS, Gas6, and TAM receptors present an exciting avenue of potential therapeutic targets against inflammation associated with atherosclerosis and cardiovascular disease.

## 1. Introduction

Advanced atherosclerosis is the cause of fatal heart attacks and strokes, which are responsible for most deaths in industrialized nations [1,2]. From cell and mouse models to clinical studies of humans, there is extensive evidence suggesting that the immune system has a decisive function in the development of atherosclerotic plaques. Atherosclerosis is characterized by non-resolving chronic inflammation of the arterial intima; both the innate and adaptive immune systems are essential in promoting and mitigating cardiovascular pathology [3].

Cardiovascular pathology begins at the level of the blood vessel. Each artery wall has three layers: (1) the inner layer or intima, (2) the media layer, and (3) the outer layer or adventitia. The intima is composed of a single layer of endothelial cells that are firmly attached to a thin basal membrane and accompanied by a subendothelial layer of collagen fibers [4]. The media layer consists of smooth muscle cells (SMCs) and a fibrous network of collagen and elastin [5]. The adventitia is a loose connective tissue that forms a sleeve around the vessel [4]. Normally, immune cells patrol the vessel wall and return to the blood circulation [6].

The retention of apolipoprotein B in the subendothelial layer of arteries leads to the oxidation of accumulated fatty acids and thus, initiates an inflammatory response in the blood vessel [7]. As the atherosclerotic plaque grows, leukocytes, primarily monocyte-derived macrophages, enter the wall and phagocytose oxidized and unoxidized cholesterol-rich lipoproteins [8,9]. The excessive ingestion of lipids transforms macrophages into foam cells that secrete extracellular matrix to trap lipoprotein and sequester proinflammatory cytokines [2,3,10]. The increase in inflammatory mediators and endothelial cell expression of adhesion molecules promotes the recruitment of additional monocytes, T cells, and neutrophils, thereby sustaining inflammatory stimuli [11]. Extracellular neutrophil traps composed of DNA and proteins appear to promote inflammation by increasing interleukin-1β production [12]. Chronic inflammation of the vessel wall is promoted by adaptive immune cells such as T and B lymphocytes. In atherosclerotic plaques, there are abnormal T helper (Th) types and two varieties of B cells, B1 and B2 cells [10,13].The B1 cells generate antibodies to intra-lesional lipids, whereas B2 cells promote the progression of disease-causing lesions [13].

Because of the release of sustained cytotoxic factors, lesional cells are unable to leave the plaque, and they become apoptotic [14]. In the early stages of atherosclerotic lesions, apoptotic cells are successfully cleared by neighboring macrophages to limit lesion cellularity. However, with time, efferocytosis fails and apoptotic cells accumulate to form a large, highly necrotic “core”, the hallmark of advanced atherosclerotic disease [14,15,16]. The necrotic nucleus of the plaque causes an imbalance in the tight plaque structure and eventually leads to dilaceration or rupture. Blood clotting then causes partial or complete vessel blockage and leads to acute thrombotic cardiovascular events, such as myocardial infarction, unstable angina, cardiac death, or stroke [14]. Thus, the clearance of dead cells is essential to prevent clinically significant atherosclerotic plaque development [14,16].

Global changes to the blood vessel occur because of attempts by the immune system to resolve the insult. The proliferation of smooth muscle cells (SMCs) surrounding the plaque thickens the vessel and the adventitia, and the vessel wall also accumulates activated immune cells. Various immune cells are normally present in the arterial wall, and progressive atherosclerosis elevates their number significantly [17]. Clinically, progressive atherosclerosis is indicated by high plasma levels of low-density lipoproteins (LDL) [18]. Advanced atherosclerosis is often present in conjunction with high blood pressure, which also activates endothelial cells because of unresolved inflammation [19]. Hypertensive conditions in blood vessels promote foam cell formation and subsequent cytokine production, enhancing atherosclerosis [20]. Figure 1 illustrates these concepts.

The TAM receptor family was originally described as a group of orphan receptors. In the 1990s, the TAM ligands were identified, namely growth arrest protein 6 (Gas6) and Protein S (PS). The ligands have different affinities for the individual TAM receptors [21,22]. Gas6 associates with all receptors; its affinity is greatest for Axl, then Tyro3, and lastly Mer. Conversely, PS does not bind to Axl and has a greater affinity for Tyro3 compared with Mer [23]. Different tissues and immune cells exhibit various levels of expression in each of these receptors. Mer is highly expressed in macrophages, whereas Axl and Tyro3 are prominently expressed in dendritic cells [24,25].

TAM receptors have essential functions in homeostasis, particularly in driving immunosuppression by inhibiting T cells and promoting efferocytosis to elicit immunosuppressive cytokines [26]. Thus, the pan-inhibition of TAM receptors is expected to remove their immunosuppressive properties and has been suggested as an alternative cancer therapy by improving antitumor immunity [27].

Both Protein S and Gas6 proteins undergo vitamin K-mediated γ-carboxylation, and they share a structural homology with a similar domain composition of approximately 42%. Despite the structural similarity, each protein has distinct functions. Protein S circulates in the plasma at ~350 nMand is involved in hemostasis, apoptosis, inflammation, and atherosclerosis [28,29]. In its hemostasis function, PS, encoded by the *PROS1* gene in humans, is predominantly expressed in the liver and secreted to mediate anticoagulation. Protein S is expressed as a signaling molecule in several other tissues [28]. In human plasma, 40% of PS exists in a free (physiologically active) form to mediate anticoagulation, and 60% of PS is bound to a C4 binding protein (C4-BP), an acute phase protein, to mediate the other physiological roles of PS [29]. Protein S contains an amino terminal γ carboxylation acid (GLA) domain, followed by a thrombin-sensitive loop region and four epidermal growth factor-like domains [30]. The C-terminal region consists of two laminin G domains that comprise the sex hormone-binding globulin domain and is sufficient for TAM receptor binding and receptor autophosphorylation [31].

Gas6 also has a thrombin-sensitive region (a disulfide-bridged thumb loop) but, unlike PS, the corresponding Gas6 region cannot be cleaved by serine proteases [32]. Gas6 circulates in the plasma at a much lower concentration of 0.25 nM and is elevated in patients with severe sepsis [33]. Unlike Protein S, Gas6 is not produced in the liver but instead, it is expressed in the lungs, kidneys, and heart. Gas6 has several functions in endothelial cells (ECs), vascular smooth muscle cells (VSMCs) and bone marrow [34]. Human platelets aggregate via TAM activation by Gas6 [35].

Defects in TAM-induced efferocytosis and the resulting inflammation promote atherosclerosis progression [36]. Recent results from the CANTOS trial (Canakinumab Anti-Inflammatory Thrombosis and Outcomes Study) have shown that interleukin-1β inhibition could reduce cardiovascular events. Although the CANTOS trial showed a significant increase in lethal infections in study participants, the CANTOS trial provided the first evidence that targeting inflammation to lower the frequency of cardiovascular events is effective regardless of a reduction in lipid levels [37]. Future therapeutic advances targeting inflammation in atherosclerosis treatment may indicate treatments that modulate TAM receptors. Herein, we discuss the recent research regarding the functions of TAM receptors and their PS and Gas6 ligands in atherosclerosis.

## 2. Discussion

### 2.1. Structure, Function, and Regulation in TAM Receptors

The Structure of TAM Receptors

The structure of TAM receptors includes an extracellular N-terminal region with two immunoglobulin (Ig)-like domains. Following the Ig-like domains, there are two fibronectin type III domains, a hydrophobic domain that crosses the cell membrane, and an intracellular tyrosine kinase C-terminal domain [38].

b.Structural Interactions between TAM Receptors, Protein S, and Gas6

Although Gas6 and PS differ in their functions, both proteins contain a 50-residue Gla-domain. The Gla-domain has a high affinity for calcium and promotes the binding of phosphatidylserine (PtdSer) molecules to the surface of platelets as well as to the outer leaf of cell membranes on apoptotic cells [39]. Structurally opposed to the Gla domain is the C-terminus of Gas6 and PS that possesses laminin G (globular) domains that facilitate interactions with the TAM receptor Ig-like domains.

The LG domains of Gas6 and PS facilitate binding to the Ig1 domains of TAM receptors to promote downstream signaling via ERK, PLC-γ, and PI3K/AKT [40,41,42]. Al Tsou et al. demonstrated that the inhibition of vitamin K-dependent γ-carboxylation within the Gla domain of Gas6 and PS prevents TAM receptor activation. Thus, the γ-carboxylation of the Gla domain may also be necessary for Gas6 and PS to activate a specific structural conformation in TAM receptors. Furthermore, the group observed the differential binding of TAM receptors to PS and Gas6. Axl and Tyro3 bind preferentially to Gas6 and PS, respectively. Gas6 can bind to Axl, Tyro3, and Mer, while PS can only bind to Tyro3 and Mer. Overall, AXL demonstrates the strongest binding to Gas6 compared to the other ligands. Mer binds with low affinity to both Gas6 and PS but increasing local concentrations of PtdSer enhances its strength of binding to Gas6 and PS. The crystal structures of TAM receptors also show that Zn^2+^, Ca^2+^, and Ni^+^ are needed for PS and Gas6 to successfully bind to and activate Tyro3 or Axl, respectively. Thus, ion interactions are important for the appropriate conformational rearrangements of TAM receptors [42]. Like other tyrosine kinase receptors, ligand binding facilitates dimerization, autophosphorylation, and coupling with proteins from different signaling pathways [43].

c.General Functions of TAM Receptors

Unlike most receptor tyrosine kinases, TAM receptors are not required for embryonic development. However, the adult expression of TAM receptors is essential. TAM receptor triple knockout (KO) mice develop common phenotypes like male infertility and blindness [44,45]. Infertility results from the inability to clear apoptotic gamete cells during spermatogenesis in the testis [46]. Similarly, retinal epithelial cells are unable to engulf the outer segment of photoreceptors and the toxic byproducts of phototransduction remain. Thus, retinal cells undergo apoptosis, leading to blindness [45].

Particularly, in atherosclerosis, TAM receptors are critically important to help clear apoptotic cells in atherosclerotic lesions. However, in advanced atherosclerosis, efferocytosis is less functional, which contributes to the post-apoptotic necrosis of lesional cells [47]. Large areas of plaque necrosis develop, making plaques susceptible to rupture or erosion [22].

Surprisingly, TAM receptors also have an important function in preventing autoimmunity [48]. Failure to clear apoptotic bodies from damaged or dying cells leads to necrosis and the development of self-antigen [49]. For example, TAM triple KO mice develop lymphoproliferative disorder from broad spectrum autoimmunity due to the chronic hyperactivation of dendritic cells (DCs), monocytes, and macrophages [50].

Likewise, the deregulation of TAM-mediated suppression strongly influences cardiovascular immunity. Under normal conditions, TAM receptors are upregulated upon innate immune cell activation to prime the system for negative feedback. After the adaptive response is initiated, increased TAM receptor ligand availability promotes the taming of the immune response [51].

After the activation of innate immune cells, ligand-mediated autophosphorylation of TAM receptors blocks the activation of the type 1 interferon receptor (IFNAR)-STAT1 complex. Normally, IFNAR-STAT1 drives inflammation by association with the R1 subunit and IFNAR, and anti-inflammatory molecular signaling begins. The transcription of suppressor of cytokine signaling (SOCS) 1 and SOCS3 proteins begins and ultimately suppresses both the cytokine receptor and Toll-Like receptor (TLR) 3, TLR4, and TLR9 pro-inflammatory pathways [52]. The downstream signaling of IFNAR-STAT1 is essential for T cells, which upon activation in T cells produces PS which acts on dendritic cell (DC) TAM receptors [53]. The deregulation of the interaction between T cells and DC in atherosclerotic lesions enhances inflammation surrounding the plaque; thus, the interruption of the interaction presents a potential therapeutic approach [54].

d.Regulation of TAM Receptors

The regulation of TAM receptors occurs at the transcriptional, post-transcriptional, and protein levels [55]. Various cytokines such as transforming growth factor beta (TGF-β) and granulocyte-macrophage colony-stimulating factors (G-CSF) upregulate the expression of Axl. Cytokines interleukin-17A and IL-10 and dexamethasone drive Mer expression [56]. Wang et al. suggested that the downregulation of Tyro3 occurs in inflammation associated with autoimmune disorders. However, little is known about cytokines that affect Tyro3 expression [57]. Other studies have reported microRNA regulation of Axl and Tyro3 in different disease pathologies. Endisha et al. reported that microRNA-34a suppresses Axl protein expression, which contributes to rheumatoid arthritis and osteoarthritis [58]. Similarly, microRNA7 inhibits Tyro3 expression and is being examined as an RNA-based therapeutic for treating aberrant Tyro3 overexpression in human colorectal carcinoma [59].

Once TAM receptors are fully synthesized and activated, their extracellular domain can be cleaved by the metalloproteinase A Disintegrin and Metalloproteinase-17 (ADAM-17). The cleavage liberates a soluble extracellular domain bound to soluble Axl or active Axl and destroys TAM receptor activity [60]. Conversely, the secretory leukocyte protease inhibitor increases the expression of Mer on the cell surface by inhibiting cleavage [61].

e.Sex differences in TAM receptor expression and coronary artery disease

Ischemic heart disease caused by coronary artery disease is the leading cause of death of women worldwide [62]. Until menopause, women have a lower risk of coronary artery disease compared to men [63]. However, decreased levels of estrogen during menopause may double the risk of coronary artery disease in women [63,64].

Previous studies report the presence of estrogen and androgen responsive elements in the Gas6 and PS promoters [65,66,67,68]. Both Gas6 and PS have a sex hormone binding globulin domain which binds to estrogen and androgen with high affinity [69,70]. Thus, sex steroid regulation of TAM receptors may occur indirectly by controlling the transcription of Gas6 and PS.

However, Salian-Mehta et al. showed that Axl sequence variants were present in males with idiopathic hypogonadotropic hypogonadism. Increases in testosterone are correlated with elevated transcribed levels of Gas6 and as testosterone declines with age, so do levels of Gas6 expression [71]. Furthermore, levels of PS expression are downregulated by estrogen and upregulated by progestins at each of the promoter sites [68,72]. The physiological significance of the sex steroid regulation of PS is under debate.

There is contradictory evidence regarding the regulation of Gas6 by sex hormones. Hung et al. found that with corresponding decreases in estrogen in post-menopausal women compared to pre-menopausal women, Gas6 levels decrease [70]. The study found that adult males have elevated levels of circulating Gas6 protein compared to females and Gas6 concentration is positively associated with sex hormone concentration and decreases with increasing age, excluding the effect of hormone therapy in both genders. Hung et al. concluded that estrogen (E2) may modulate the Gas6 and subsequently the Gas6/TAM receptor system for diverse clinical consequences in women [73]. However, Holden et al. showed that women with coronary artery disease, which is more likely to manifest after menopause, had elevated plasma levels of Gas6 compared to men with CAD. The study suggested, contrary to Hung et al. that lower levels of estrogen post-menopause could be correlated with higher levels of Gas6 and thus, modify cardiovascular risk [74].

Furthermore, elevated levels of Gas6 have been associated with obesity, markers of endothelial dysfunction, and atherogenesis in women only [63,74,75]. Both Gas6 and estrogen impact common downstream pathways controlling glucose and lipid homeostasis, as well as inflammation and apoptosis signaling [63]. Sola et al. found that levels of Gas6 and soluble Axl (sAxl) have also been detected in obese adolescents compared to adolescents of normal body mass index (BMI). When the study was conducted in adult women and men, this association was only significant in women. However, for the same BMI, women tend to have, on average, 10% greater body fat than men and in morbidly obese women, visceral fat is significantly elevated [75]. The relationship between BMI and Gas6 levels may be elevated in women due to increased visceral adipose tissue. Obesity is an independent risk factor for cardiovascular disease, and further investigation into the association between Gas6, BMI, and coronary artery disease is needed [75,76].

Nanao-Hamai et al. stated that Gas6 and estrogen may have cardioprotective effects on the heart in women. Estrogen (E2), via the estrogen receptor α (ERα), stimulates Gas6 transcription to inhibit vascular smooth muscle cell (VSMC) apoptosis and calcification, advanced atherosclerosis, and vascular calcification. Thus, the interaction between Gas6 and estrogen may be cardioprotective [64].

However, the synergy of Gas6 and estrogen effects on insulin resistance is debated. Kuo et al. stated that the estrogen-mediated estrogen-responsive element in Gas6 activates the PI3K/Akt pathway leading to increased insulin sensitivity [63]. However, elevated levels of Gas6 have been associated with insulin resistance [34,74]. The direct role of estrogen and Gas6 on coronary artery disease has not yet been deciphered and more investigation is needed about the tissue-specific regulation of Gas6, PS, and the TAM system via estrogen and androgen response elements [63,73].

### 2.2. The Role of Axl in Atherosclerosis and Cardiovascular Disease

Gas6-Axl and regulation of cardiovascular remodeling

Vascular remodeling involves structural changes to the vascular wall in response to injury or inflammatory mediators. Chronic conditions such as hypertension or atherosclerosis drive vascular remodeling [77]. Endothelial cells (ECs), vascular smooth muscle cells (VSMCs), fibroblasts, and myofibroblasts contribute to vascular remodeling by four main processes: (1) cellular migration, (2) proliferation, (3) survival, and (4) modification of extracellular matrix. In each of these processes, Axl and its ligands have key functions [78].

Protein S and Gas6 are secreted by, and drive the proliferation of, vascular smooth muscle cells. Melaragno et al. showed that, after balloon injury in a rat carotid artery, Axl expression was increased in VSMCs and the upregulation of Gas6 and Axl was temporally correlated with neointima formation [79]. In Axl double KO mice, Chen et al. found that vessel injury resulted in reduced intimal thickening compared with control mice, and Axl deficiency led to reduced systolic blood pressure and reduced modeling of the mesenteric artery in a mouse model of hypertension [80]. We need further understanding of the function of Gas6-Axl signaling in VSMC proliferation and migration.

In hypertension, Angiotensin II (Ang II) is produced as a byproduct of the renin-angiotensin system. Angiotensin II has an important function in several cardiovascular pathologies, including vascular remodeling and neointima formation [81]. Interestingly, Ang II also activates the Gas6-Axl pathway and is needed for the effects of Axl on VSMC proliferation [82]. Likewise, reactive oxygen species (ROS) elicit pathological effects on vasculature partially through Axl expression in VSMCs [83]. By inducing interactions between Axl glutathiolated non-muscle myosin heavy chain (MHC)-IIB, ROS increases VSMC migration in vascular injury. In targeting Axl in vitro, oxidative stress significantly decreases and reduces VSMC migration [84]. Lee et al. found that Axl deficiency showed an increase in VSMC apoptosis and reduces intima–media thickening following arterial ligation, suggesting that Axl/Gas6 may prolong VSMC survival [85].

After vascular injury, VSMCs proliferate and migrate, aggravating vascular disease. Although the molecular mechanisms of neointima formation are not fully understood, Lee et al. reported that extracellular vesicles (EVs) are mediators of intercellular communication that aids neointima formation [86]. McShane et al. showed that small extracellular vesicles increase following the ligation of the right carotid artery in rats, and subsequently neointima formation is aggravated via the interaction with membrane PtdSer [23]. Furthermore, the incubation of damaged vessels with EVs enhanced Axl and Mer phosphorylation in VSMCs and increased downstream signaling pathways associated with Akt, extracellular signal-regulated kinase (ERK), and focal adhesion kinase (FAK) [86]. Okamoto et al. reported that the transcription factor YAP increases the expression of Axl, constituting an Axl-positive feedback loop [87]. Small molecule inhibitors of both Axl and Mer hindered VSMC proliferation and migration [88]. This evidence shows the potential of Axl inhibitors to prevent the aggravation of vascular injury associated with atherosclerosis.

Similarly, Liu et al. showed that Axl-Gas6 signaling promoted mitogenic effects on fibroblasts, protecting cells from apoptosis [89]. Signaling also affects the survival and migration of endothelial cells through the upregulation of vascular endothelial growth factor A [90]. Axl also directly regulates cytokine/chemokine expression and ECM remodeling in the vessel wall. Ultimately, Axl mediates these functions through the activation of phosphatidylinositol-3-OH kinase (PI3K)/protein kinase B (Akt), sarcoma (SRC) signaling pathways and extracellular signal-regulated kinases (ERKs), like other receptor tyrosine kinase-mediated processes [91].

The excessive stretching of endothelial cells releases Gas6 which activates Axl on monocytes and promotes inflammation [92,93]. With a mouse model of hypertension, Chen et al. demonstrated that brief hypertensive episodes induced chronic aortic remodeling consistent with persistent low-grade inflammation of the aorta and kidneys. Furthermore, the blockade of Axl and the subsequent blockade of Gas/Axl signaling lessened the severity of hypertension [80].

Interestingly, glucose concentration affects Axl behavior in vitro. Zdzalik-Bielecka et al. exposed VSMCs to a low glucose concentration (5.5 mmol/L) and a high glucose concentration (27.5 mmol/L). They found that Axl preferentially interacted with proteins in the PI3K signaling pathway in the low-glucose condition [94]. In low-glucose conditions, Axl stimulates anti-apoptotic signaling and enhances the survival of VSMCs. In contrast, Axl is associated with the signaling proteins of the ERK1/2 pathway in high glucose conditions, driving VSMC migration. This finding may have important implications in our understanding of the metabolic syndrome [95].

b.Gas6-Axl regulation of vascular calcification

Vascular calcification is another process that enhances inflammation associated with advanced atherosclerosis. Calcium build-up in the vasculature correlates with worse clinical outcomes [96]. However, in the absence of calcification in coronary arteries, individuals have a low risk of cardiovascular disease even if they have high levels of traditional risk factors [97]. Vascular pericytes undergo osteogenic differentiation during calcification and in vitro Axl is downregulated during this osteogenic differentiation [98]. Levels of Axl also decrease when cultured VSMCs calcify their matrix, and Axl overexpression prevents calcification [99]. Notably, Badi et al. showed that miR-34a promotes VSMC calcification, and this calcification correlated with decreased Axl expression in cultured VSMCs [100]. Interestingly, Son et al. found that hydroxy-3-methylglutaryl coenzyme A reductase inhibitors (statins) prevent phosphate-induced calcification by VSMCs in vitro via the restoration of the Gas6-Axl mediated survival pathway [101]. It is unknown whether Axl affects vascular calcification in vivo.

c.Gas6-Axl and inflammation in cardiovascular disease

High levels of vasculature inflammation promote cardiovascular pathology. Although there are no studies of the impact of Axl on atherosclerosis pathology, Gerloff et al. found that Axl expression was downregulated in atherosclerotic plaques compared with normal carotid arteries [102]. Interestingly, Axl expression is higher in vessels such as the left internal mammary artery, which is less prone to developing atherosclerosis compared with vessels more prone to developing atherosclerosis, such as the aorta [103]. Conversely, Liu et al. detected elevated levels of soluble Axl in the acute coronary syndrome, and Gas6 expression increases in ECs, VSMCs, and macrophages associated with atherosclerosis development [104]. Some groups suggest that the genetic knockout of Gas6 increases plaque stability although in clinical studies, Sunbul et al. found that the levels of Gas6 correlated with the increased risk of coronary artery disease in patients with psoriasis [105]. However, recently, You et al. showed that Axl signaling dampened dendritic cell maturation when exposed to cholesterol and Axl may be a target to prevent the stimulation of dendritic cells [106]. Axl is also necessary for the survival of T lymphocytes, and Axl influences vascular remodeling and inflammation in a mouse model of hypertension. Axl drives the pro-inflammatory activation of VSMCs in vein graft remodeling. The depletion of Axl in the vein graft donor and recipient mouse leads to decreased levels of Axl, less immune activation, and the subsequent downregulation of various pro-inflammatory cytokines [107]. Axl deficiency increases the expression of SOCS1 in VSMC, contrary to what is observed in immune cells [23,107].

Axl also has a function in hypertension [108]. Axl-expressing immune cells drive pro-inflammatory gene expression and increase immune cell infiltration in the kidneys at early stages of hypertension and was detrimental in late stages of hypertensive disease [109]. Li et al. showed, in a recent clinical study, that decreased serum concentrations of Gas6, Axl, and soluble Axl were associated with high HbA1c values and could predict the severity of diabetic nephropathy, which is often associated with hypertension [110].

In patients with heart failure and post ST-segment elevation myocardial infarction, Axl and soluble Axl levels are elevated in cardiac tissue [111]. Axl level is predictive of adverse pathology, such as the extent of left ventricle remodeling and future cardiovascular events [112,113]. Decreased hypertrophy, fibrosis, and contractile dysfunction from chronic stress induced by aortic banding was demonstrated in Gas6 deficient-mice, and cardiac-specific Gas6 overexpression enhanced these pathologies. Gas6 acts through the ERK pathway to drive cardiac hypertrophy, and the reversal of hypertrophy was shown with ERK inhibitor treatment [82]. Hence, the Gas6-Axl interaction might be a therapeutic target in heart failure.

d.The role of Axl and Mer in Sterile Wound Healing

Axl and Mer are necessary for macrophage activation following sterile wound healing associated with myocardial infarction [114]. In myocardial ischemia/reperfusion infarction, the cross signaling between Axl and TLR4 in cardiac macrophages directed a switch from glycolysis to lipid metabolism and the secretion of proinflammatory IL-1β [114,115]. This maladaptive condition promoted increased intramyocardial inflammation, impaired contractile function, and adverse ventricular remodeling [114]. Although Mer is cardioprotective, Axl functions independently to reduce the efficacy of cardiac repair. However, small molecule inhibitors of Axl and Mer cleavage significantly improved cardiac healing [23]. Figure 2 summarizes the key roles of Axl-Gas6 in cardiovascular disease.

### 2.3. The Function of Mer in Atherosclerosis and Cardiovascular Disease

Mer-mediated resolution of inflammation and efferocytosis in atherosclerosis

The resolution of inflammation is a highly coordinated process that counterbalances excess inflammation without compromising host defense to restore tissue homeostasis after injury [116]. When inflammation fails to resolve, it exacerbates numerous chronic inflammatory diseases, including atherosclerosis [117].

Senescent cells release factors that promote Mer cleavage, thus promoting impaired efferocytosis. However, specialized pro-resolving mediator (SPM) resolvin 1 (RvD1) prevents Mer cleavage [118]. By downstream signaling, Mer promotes SPM production to resolve inflammation [119]. Specialized pro-resolving mediators (SPMs) include lipoxins, protectins, maresins like MaR1, and resolvins such as resolvin 1 and are crucial anti-inflammatory lipid mediators. Lipoxygenase enzymes, such as 5-lipoxygenase and 12/15-lipoxygenase, convert arachidonic acid to lipoxins, such as LX4A and docosahexanoic acid, to form resolvins, such as RvD1 [120]. Upon release, SPMs block inflammatory cell influx and promote the emigration of inflammatory cells to limit tissue damage and begin the process of healing [121]. In advanced atherosclerosis, pro-inflammatory mediators, such as leukotrienes, predominate over pro-resolving mediators. Consequently, the ratio of leukotrienes to SPMs is significantly decreased in advanced atherosclerotic plaques in humans and mice [122].

Without effective resolution, plaques become clinically dangerous. The lack of effective efferocytosis in association with damage-associated molecular pattern (DAMP)-mediated inflammation promotes the formation of a necrotic core and the thinning of the protective collagen cap that overlies the core [123]. Evidence suggests that SPMs decrease plaque necrosis and inflammation [124]. For example, the administration of RvD1 decreased the size of necrotic cores, improved intralesional efferocytosis and thickened collagenous fibrous caps in a mouse model of atherosclerosis. Consequently, lesion size decreased, and platelet and leukocyte influx lessened [125]. When treated simultaneously with RdV1 and maresin MaR1, plaque stability prevailed by decreasing lesional macrophage numbers, thickening collagen caps, and halting necrotic core growth [126]. Maresin-1 prevents atherosclerotic progression by suppressing necrosis and limiting macrophage numbers, while simultaneously increasing the fibrotic strength of the smooth muscle coat [127].

Unlike the contradictory functions of Axl in promoting and inhibiting cardiovascular pathology, Mer-mediated efferocytosis and inflammation resolution has a protective function in atherosclerosis by limiting the formation of an inflammatory necrotic core [128]. Liao et al. showed that Mer KO mice had a decreased clearance of apoptotic bodies. Consequently, plaque necrosis and inflammation were accelerated, leading to advanced atherosclerotic lesions [36]. Because of the downregulation of Mer in atherosclerotic plaques, atherosclerosis becomes more severe. Mer-mediated efferocytosis in atherosclerosis is prevented by macrophages that express calcium/calmodulin-dependent protein kinase II gamma (CaMKII-gamma). CaMKII-gamma prevents Mer expression by inhibiting transcription factors ATF6 and LXR-α [129]. Similarly, the immunoproteasome subunit β-5i decreases Mer expression in macrophages and enhances the necrotic core area in atherosclerotic lesions [130]. Patel et al. suggested that C-C chemokine receptor type 2 expressed by infiltrating circulatory monocytes negatively affects the ability of Mer to drive phagocytic repair following ischemia-reperfusion [131].

When Mer-mediated efferocytosis functions in an atherosclerotic plaque, Mer binds apoptotic cells by way of bridging molecules Gas6 or PS and mediates the uptake of apoptotic debris via actin signaling pathways [132]. Unlike Axl, Mer is the dominant efferocytosis receptor in the lesional clearance of apoptotic debris, whereas Axl expressed on dendritic cells does not affect efferocytosis [133]. Mer deficiency enhances pathology in mouse models of atherosclerosis, showing a marked decrease in apoptotic body clearance and the subsequent development of large necrotic plaque lesions [134].

Efferocytosis via the Mer receptor stimulates the formation of specialized pro-resolving mediators (SPM) [135]. The activation by apoptotic cells or Mer ligands, such as Gas6 and Protein S, promotes the cytoplasmic localization of non-phosphorylated 5-lipoxygenase in macrophages. Upon the activation of ERK in macrophages, the expression of sarcoplasmic/endoplasmic reticulum calcium ATPase 2 increases, consequently decreasing the cytosolic calcium concentration and suppressing CaMKII activity [135,136]. In turn, decreased cytosolic calcium reduces p38 MAP kinase and MAPKAP kinases MK2 activity. Thus, the unphosphorylated cytoplasmic form of 5-lipoxygenase predominates and enhances SPM biosynthesis [135]. The release of SPMs resolves inflammation in the environment surrounding apoptotic and necrotic cells [137]. Mer deficiency in sterile peritonitis and ischemia-reperfusion injury delays healing by suppressing the synthesis of SPMs such as lipoxins and resolvins [138,139].

In the development of advanced atherosclerotic lesions, macrophage Mer deficiency occurs near the necrotic cores of human plaques because of ADAM17-mediated Mer cleavage. Mer is cleaved at proline-485, and cells that express Mer lacking residues 483-488 are resistant to ADAM17 cleavage [140]. ADAM17 is activated by the products of polyunsaturated fatty acid oxidation and by inflammatory mediators trapped in the necrotic core. The presence of inflammatory stimuli promotes the ADAM17 cleavage of the Mer ectodomain [141]. Levels of the resulting soluble Mer are elevated in murine models of atherosclerosis and in patients experiencing symptomatic atherosclerotic plaques in the carotid artery. Cai et al. showed that soluble Mer, which is the inactive version of the Mer receptor, is a marker of defective Mer-mediated efferocytosis and is associated with advanced plaque progression [129].

Kawai et al. showed that macrophages near the necrotic core of an atheroma have a lower expression of Mer and a higher expression of ADAM17 compared with macrophages on the periphery of atherosclerotic lesions [142]. Furthermore, lower levels of soluble Mer in individual plaques were correlated with worse necrosis, and mice resistant to Mer cleavage showed improvement in lesional efferocytosis, stable plaque formation and inflammation resolution [143]. Thus, in advanced atherosclerosis, Mer cleavage may contribute to defective efferocytosis and failed inflammation resolution.

Remarkably, cleavage-resistant Mer in the presence of elevated ADAM17 is protective in different mouse models [140]. In an ischemia-reperfusion-induced lung injury model and the sterile peritonitis model, cleavage-resistant Mer promoted efferocytosis, inflammation resolution, and high circulating levels of SPMs [138]. An atherosclerosis model demonstrated lesions with improved efferocytosis, thicker fibrous caps, increased pro-resolving lipid mediators, and smaller necrotic cores. After myocardial ischemia-reperfusion injury, these mice also had improved efferocytosis, reduced overall infarct size, and improved cardiac function [129,135]. Sufit et al. presented evidence that cleavage-resistant Mer promoted pro-resolving mediators and restored senescence-reduced efferocytosis in models of aged mice, potentially impacting aging [144]. Conversely, Mer KO mice had an inherent increased production of pro-inflammatory cytokines such as IFN-γ, IL-12, and tumor necrosis factor α [145]. Consequently, anti-atherogenic IL-10 production is downregulated, demonstrating the function of Mer in dampening inflammation [140].

b.Axl regulation of Mer-mediated Efferocytosis in Cardiovascular Disease

The regulation of Mer-mediated efferocytosis may also arise from soluble Axl. Weinger et al. showed that, as levels of soluble Axl rise in association with Gas6 in mouse serum, soluble Axl may neutralize Gas6 in serum. By blocking the interaction of Gas6 with TAM receptors, particularly to Mer, soluble Axl indirectly decreases the interaction between Mer and apoptotic cells and limits efferocytosis. Thus, soluble Axl may influence atherosclerosis pathology by a soluble Axl/Gas6/Mer axis [146].

Recently, Wu et al. demonstrated that, in vessels with disturbed flow characteristic of atherosclerosis, the absence of Mer resulted in endothelial and mitochondrial dysfunction in the human aorta. The absence of Mer also induced abnormal endothelial thickening associated with decreased endothelial efferocytosis, and Mer absence promoted the development of atherosclerosis [147]. Thus, a strategy to regulate Mer in EC efferocytosis may be of therapeutic promise in atherosclerosis.

c.Protein S and Mer promote the development of Atherosclerosis

The relation between Mer and PS also has an important function in the modulation of atheroma development [69]. Because PS from plasma accumulates in the necrotic core of a developed plaque, PS interacts with negatively charged phospholipids on the membranes of dying cells. Through the interaction with Mer, PS inhibits acetylated LDL uptake by macrophages and downregulates scavenger receptor A, which is mainly expressed by macrophages [148]. Thus, PS could have a dual function in advanced lesions to prevent inflammation. Although PS blocks LDL phagocytosis, PS promotes Mer-mediated efferocytosis of apoptotic cells [69].

Protein S-Mer may participate in atherosclerosis by preventing LDL phagocytosis and inhibiting inflammation. Khatana et al. reported that, in the presence of excess lipoprotein, LXR expression is activated, upregulating Mer and mitigating pro-inflammatory cytokine release upon the engagement of cholesterol-loaded macrophages [149]. By way of Mer, PS also blocks the expression of macrophage scavenger receptor A and reduces the uptake of modified lipoproteins [136]. Thus, Mer and PS are crucial for the attenuation of inflammation in atherosclerosis [69,150]. Figure 3 summarizes these concepts.

d.Mer-mediated efferocytosis in myocardial infarction

Following myocardial infarction (MI), an essential first step in the healing process is the clearance of dead cardiomyocytes by several types of immune cells. DeBerge et al. showed that Mer-expressing monocytes/macrophages are needed to clear injured cardiomyocytes and improve cardiac tissue remodeling after MI in mice [151]. After an experimental MI, Mer KO mice experienced the accumulation of apoptotic cardiac cells, resulting in larger infarct sizes and worse cardiac functional outcomes [152]. During homeostasis, cardiomyocytes release nonfunctional mitochondria, known as exophers, in subcellular particles [153]. Exophers are engulfed and cleared by macrophages in cardiac tissue, preventing inflammation and phagocytosis by the surrounding cardiomyocytes and maintaining cardiac homeostasis [154]. When Mer is ablated on cardiac macrophages, metabolic function is impaired due to residual exophers in cardiac tissue [155].

Neutrophils are also critical for influencing Mer expression on cardiac macrophages. The secretion of neutrophil gelatinase-associated lipocalin (NGAL) activates Mer on cardiac macrophages to facilitate efferocytosis. The absence of NGAL reduces expression of Mer on the cardiac macrophage membrane, thereby impairing efferocytosis [156].

Dying cardiomyocytes induce the expression of Mer on their surface, which prevents their engulfment. Subsequently, macrophages upregulate CD47, a “don’t eat me” marker for dying cells that prevents efferocytosis [157]. Cell marker CD47 is often found on lingering dead and necroptotic cells [158]. The treatment with an anti-CD47 antibody leads to an increase in the levels of SPMs, such as RvD1 in plaques, and suggests another feedforward circuit to inhibit inflammation [143]. Although residual necrotic cardiomyocytes in MI is pathological and may lead to worsening clinical outcomes, preventing efferocytosis may preserve cardiomyocyte numbers in lieu of low regenerative capacity [22]. The treatment of mice with anti-CD47 enhanced cardiomyocyte phagocytosis and decreased infarct size, leading to better post-MI outcomes [143]. Importantly, anti-CD47 treatment may be a therapeutic approach focused on pro-resolving mechanisms instead of the direct inhibition of inflammation.

### 2.4. Tyro3 in Cardiovascular Disease

Because of the low levels of expression of Tyro3 in the vasculature compared with the central nervous system and the reproductive system, the function of Tyro3 in cardiovascular disease has not been studied in detail [159]. Hurtado et al. showed the association of Tyro3 and Mer variants with carotid atherosclerosis [160].

However, Tyro3 is thought to suppress a subset of CD11c+ DC that expresses programmed cell death protein 2 and decreases Th2-associated molecule production [161]. Part of a negative feedback loop, IL-4 increases the expression of Protein S in T cells and then activated Tyro3-mediated suppression of Th2 activation. Although IL-4 and IL-5 protect against atherosclerosis, IL-9 may promote atherosclerosis [162]. Furthermore, the fibrosis of heart tissue in the process of aging and reperfusion injury is promoted by IL-13 secreted by regulatory T cells, which may also promote efferocytosis in atherosclerotic lesions [23,163]. Further studies are needed to elucidate the function of Tyro3 in cardiovascular disease.

## 3. Conclusions

The TAM receptors, Gas6, and PS have many functions that suppress or drive cardiovascular pathology. There is no consensus to suggest that a detrimental function or protective effect of Axl-Gas6 stems from differences in the activation of inflammation to signaling to suppress the immune response. Although Axl may protect against vascular calcification, the function of Axl-Gas6 in taming inflammation in vascular cells is unclear. However, Mer-PS is fundamentally protective in its function against cardiovascular disease. From immune response suppression to immune response resolution, Mer is instrumental in preventing the inflammation characteristic of atherosclerotic lesions. The protective property of Mer in cardiac tissue presents a new therapeutic consideration for heart disease. Of the TAM receptors, Tyro3 remains the least studied in cardiovascular disease, and the few studies that exist suggest that Tyro3 could be a therapeutic target after myocardial injury. In sum, the functions of TAM receptors, Gas6, and PS present an exciting opportunity of therapeutic interventions in the cardiovascular setting.

## 4. Future Directions

The high mortality and morbidity associated with advanced atherosclerosis necessitates a broader understanding of novel therapeutic targets to treat the disease that can develop in the absence of major clinical symptoms. Although representing a broad range of physiological activities, the functions of Tyro3, Axl, and Mer receptors may be critically important to target in the development of novel therapeutics for the treatment of cardiovascular disease.

## Figures and Tables

**Figure 1 ijms-25-12736-f001:**
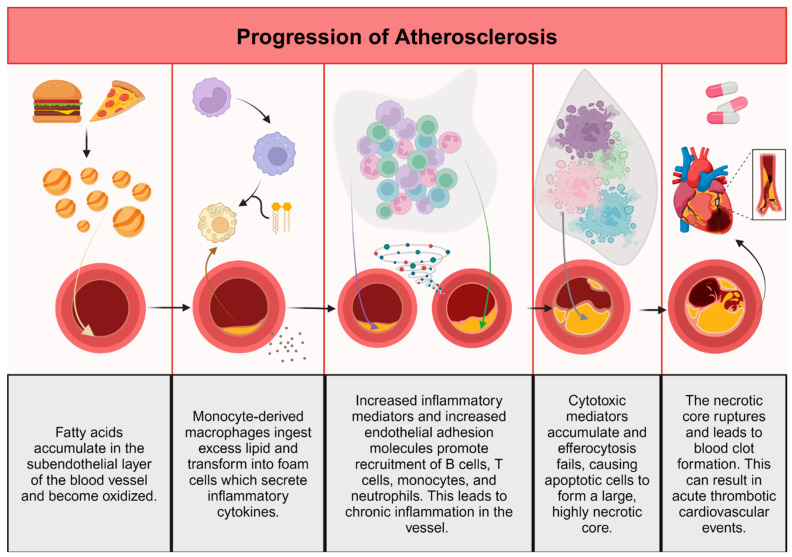
Pathogenesis of atherosclerosis. Created with BioRender.com.

**Figure 2 ijms-25-12736-f002:**
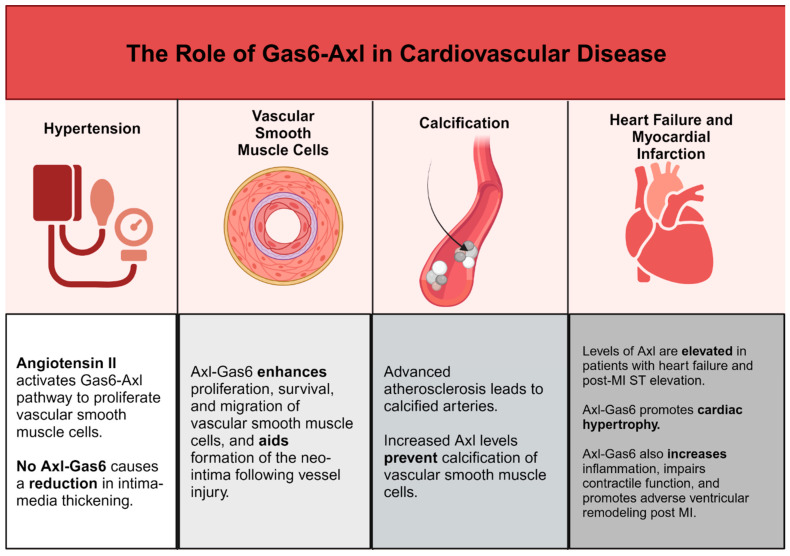
Summary of the functions of Gas6-Axl in cardiovascular disease. Created with BioRender.com.

**Figure 3 ijms-25-12736-f003:**
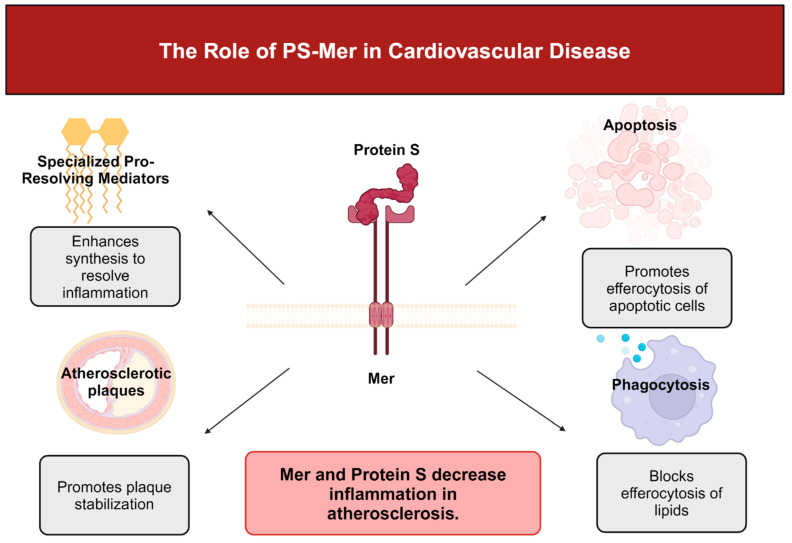
Summary of the functions of PS-Mer in cardiovascular disease. Created with BioRender.com.

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
