# Peer review of "Functions of TAM Receptors and Ligands Protein S and Gas6 in Atherosclerosis and Cardiovascular Disease"

_ijms, 2024, doi:10.3390/ijms252312736_

Round 1

Reviewer 1 Report

Comments and Suggestions for Authors

In the present review Prouse et al. provide insights into the role of TAM receptors and its ligands (Protein S and Gas6) in vascular pathophysiology associated with atherosclerosis / cardiovascular disease. The review highlights a unique pathway that may play a critical role in regulating vascular remodeling in CVD. The review is informative and systematically describes the role of the various components (TAM receptors and its ligands) within the vasculature and CVD. The references cited are appropriate for the review. Overall the review provide a good case for a protective role of TAM receptors, Protein S and Gas6 against multiple vascular abnormalities linked to atherosclerosis / CVD. I have a few suggestions that the authors may consider incorporation.

Comments:

1: The authors have not addressed the issue of gender based differences on the impact of TAM and its ligands. Since coronary artery disease is the leadig cause of mortality in women a section on what is known about the impact of this pathway and whether there are gender based differences would be important. The authors should consider incorporating it.

2: Are TAM expression or level of its ligands modulated by sex steroids ? some information and discussion would make it relevant to a wider audience. If possible, the authors should add this. 

Author Response

Reviewer 1

In the present review Prouse et al. provide insights into the role of TAM receptors and its ligands (Protein S and Gas6) in vascular pathophysiology associated with atherosclerosis / cardiovascular disease. The review highlights a unique pathway that may play a critical role in regulating vascular remodeling in CVD. The review is informative and systematically describes the role of the various components (TAM receptors and its ligands) within the vasculature and CVD. The references cited are appropriate for the review. Overall the review provide a good case for a protective role of TAM receptors, Protein S and Gas6 against multiple vascular abnormalities linked to atherosclerosis / CVD. I have a few suggestions that the authors may consider incorporation.

Comments:

  1. The authors have not addressed the issue of gender-based differences on the impact of TAM and its ligands. Since coronary artery disease is the leading cause of mortality in women a section on what is known about the impact of this pathway and whether there are gender-based differences would be important. The authors should consider incorporating it.

We are grateful for this insight. We conducted another literature search, and we have found no recent studies directly investigating the role of TAM receptors in coronary artery disease associated with women. The studies we have found evaluated the role of Gas6 in risk factors associated with coronary artery disease, such as obesity and insulin resistance. We have incorporated these findings in our discussion beginning on page 12.

  1. Are TAM expression or level of its ligands modulated by sex steroids? some information and discussion would make it relevant to a wider audience. If possible, the authors should add this. 

Thank you for the suggested addition to the manuscript. We have conducted a literature search, and we found that few studies investigated how sex steroids directly impact expression levels of TAM receptors. TAM receptors are necessary for male fertility and spermatogenesis, which notes the androgen-responsiveness of TAM receptors via their sex-hormone binding globulin (SHBG) domain. Although a brief statement has summarized this role, further information has been provided on page 12.  There is little information on how estrogen impacts levels of TAM receptor expression, but studies have investigated how estrogen indirectly affects TAM receptor expression through modulating levels of Gas6 and PS. Thus, the indirect impact of estrogen on TAM receptors via common ligands has been studied but no direct physiological mechanisms have yet been published. All the currently available information about the role of estrogen in TAM receptor, Gas6, and PS regulation has also been included on page 12.

Reviewer 2 Report

Comments and Suggestions for Authors

A very enjoyable manuscript, my compliments to the authors.

The technical content is well presented and thorough.

A few suggestions:

1. Please reorganize the paper to present a more cohesive flow of topics and sections, some appear mislabeled and to enhance the strength of the paper, i would advise organizing around the central themes of your paper.

For instance, under TAM receptors, subsections should be a.,b.,c. etc

From TAM receptors, would have a bridge paragraph describing the relationship between these and the subsequent ligands, before discussing each ligand with well labeled sub sections.

2. concluding statement should preferably read "Although representing a broad range of phys-ological activities, the functions of Tyro3, Axl, and Mer receptors may be critically important to ""target in development of novel therapeutics for the treatment"" of cardiovascular disease.

Author Response

Reviewer 2

A very enjoyable manuscript, my compliments to the authors.

The technical content is well presented and thorough.

A few suggestions:

  1. Please reorganize the paper to present a more cohesive flow of topics and sections, some appear mislabeled and to enhance the strength of the paper, I would advise organizing around the central themes of your paper.

For instance, under TAM receptors, subsections should be a.,b.,c. etc

From TAM receptors, would have a bridge paragraph describing the relationship between these and the subsequent ligands, before discussing each ligand with well labeled sub sections.

Thank you for your advice. We have reorganized the manuscript appropriately and have divided and organized the information in a more cohesive manner. Changes in the headings for each section were made to label each sub-section more appropriately.

We also added the bridge paragraph on page 8 of the manuscript and added an additional section to the manuscript which discusses the structural interactions between PS, Gas6, and TAM receptor ligands. By adding this section, the readers will have foundational knowledge necessary for understanding why certain ligands only bind to certain receptors for the rest of the paper.

  1. Concluding statement should preferably read "Although representing a broad range of physiological activities, the functions of Tyro3, Axl, and Mer receptors may be critically important to ""target in development of novel therapeutics for the treatment"" of cardiovascular disease.

Thank you for your suggestion. We have corrected the statement as advised.